# Gut Microbiome in Non-Alcoholic Fatty Liver Disease: From Mechanisms to Therapeutic Role

**DOI:** 10.3390/biomedicines10030550

**Published:** 2022-02-25

**Authors:** Haripriya Gupta, Byeong-Hyun Min, Raja Ganesan, Yoseph Asmelash Gebru, Satya Priya Sharma, Eunju Park, Sung-Min Won, Jin-Ju Jeong, Su-Been Lee, Min-Gi Cha, Goo-Hyun Kwon, Min-Kyo Jeong, Ji-Ye Hyun, Jung-A. Eom, Hee-Jin Park, Sang-Jun Yoon, Mi-Ran Choi, Dong-Joon Kim, Ki-Tae Suk

**Affiliations:** Institute for Liver and Digestive Diseases, College of Medicine, Hallym University, Chuncheon 24252, Korea; haripriya.gupta@hallym.ac.kr (H.G.); 43568@hallym.ac.kr (B.-H.M.); rg@hallym.ac.kr (R.G.); yoseph@hallym.ac.kr (Y.A.G.); satyapriya83@hallym.ac.kr (S.P.S.); epark312@hallym.ac.kr (E.P.); lionbanana@hallym.ac.kr (S.-M.W.); jj_jeong@hallym.ac.kr (J.-J.J.); qlstn5549@gmail.com (S.-B.L.); qjarlf987@naver.com (M.-G.C.); ninetjd@naver.com (G.-H.K.); 43916@hallym.ac.kr (M.-K.J.); 43566@hallym.ac.kr (J.-Y.H.); 43563@hallym.ac.kr (J.-A.E.); 43565@hallym.ac.kr (H.-J.P.); ysjtlhuman@gmail.com (S.-J.Y.); choimr@hallym.ac.kr (M.-R.C.); djkim@hallym.ac.kr (D.-J.K.)

**Keywords:** microbiota, NAFLD, microbial functions, microbiota-based approach

## Abstract

Non-alcoholic fatty liver disease (NAFLD) is considered to be a significant health threat globally, and has attracted growing concern in the research field of liver diseases. NAFLD comprises multifarious fatty degenerative disorders in the liver, including simple steatosis, steatohepatitis and fibrosis. The fundamental pathophysiology of NAFLD is complex and multifactor-driven. In addition to viruses, metabolic syndrome and alcohol, evidence has recently indicated that the microbiome is related to the development and progression of NAFLD. In this review, we summarize the possible microbiota-based therapeutic approaches and highlight the importance of establishing the diagnosis of NAFLD through the different spectra of the disease via the gut–liver axis.

## 1. Introduction

The global burden of non-alcoholic fatty liver disease (NAFLD) is growing at an alarmingly rate, and it has an estimated prevalence of 24–45% [1,2]. Along with escalating morbidity, the global mortality among the NAFLD population is also an important concern, with NAFLD deaths accounting for 23% to 29% of total deaths, and the ratio is expected to be higher in 2030 [3]. NAFLD progression could develop into more sever conditions, such as fibrosis, and lead to life-threatening states, including cirrhosis and hepatocellular carcinoma (HCC) [4,5]. Mortality and liver-associated complications start to become aggravating in the fibrosis stage in NAFLD patients [6]. The advanced stages of NAFLD are common comorbidities of metabolic syndrome and are a great affliction for mankind [3].

Despite two decades of research into establishing the link between known pathophysiological pathways and NAFLD progression, the underlying factors remain elusive [7]. The physiological pathways which are in distresses and known to be crucial in NAFLD generation follow the sequence of initially impaired metabolic functions for facilitating fat storage in hepatocytes, which leads to steatosis and lipotoxicity. Steatosis increases oxidative stress and mitochondrial disfunctions, which causes constant hepatocytic injuries, causing the activation of inflammatory cascades. This elicits responses to non-parenchymal liver cells, such as the activation of hepatic stellate cells and the infiltration of immune cells, resulting in non-alcoholic steatohepatitis (NASH) [8]. The series of pathophysiological events are well known, but the intra- and inter-connectivity of these events cooccurrence is not. In addition to intrahepatic etiology, numerous other mechanisms have also been proposed to be potential pathophysiological mechanisms for NAFLD generation. Other etiologies such as the crosstalk amongst dysfunctional organs, including the adipose tissue, brain and gut, caused by metabolic complications, age-related factors or genetic susceptibility, is also involved in the pathophysiology of NAFLD [7,9]. Intriguingly, exploring the interactions between the gut and liver has become important in gaining a better and more in-depth understanding of the gut–liver axis in NAFLD.

Given the known multifactorial drivers in the pathogenesis of NAFLD, the “one size fits all” treatment methodology for NAFLD may not be ideal [10]. Furthermore, the initial stages of the mechanism involved in the development of NAFLD are not studied at a clinical trial level, since most of the clinical studies are focused on NASH and more advanced stage to severe cirrhosis [11]. Hence, a comprehensive knowledge of not only the advanced stages, but the full spectrum of the disease may be essential to provide valuable new insights into the disease and its progression. Additionally, the growing body of evidence suggests that functional and compositional changes in the gut microbiota contribute and promote the progression of NAFLD. This alteration of gut microbiota starts in the early phase of the disease and has an impact along the spectrum of disease conditions [12]. Given the prospect of integrating the gut microbiota into the strategies for therapeutic interventions for NAFLD, this review aims to explore the possible microbiota-based therapeutic approaches and increase the understanding of the importance of establishing the diagnosis of NAFLD through different spectra of the disease via the gut–liver axis.

## 2. Microbial Pathophysiology Associated with Non-Alcoholic Liver Disease

The liver is directly connected anatomically and physiologically with the gut through the hepatic portal vein. This connection facilitates bidirectional communications between the liver and its by-products and the gut microbiota and its metabolites [13]. NAFLD is strongly linked with metabolic syndrome and shares the common pathways involving obesity, type 2 diabetes mellitus, insulin resistance, hyperlipidemia and atherosclerosis [14]. From various pre-clinical models, it was concluded that a high fat/calorie diet alters the gut microbiome, causing dysbiosis, which successively ruptures the intestinal barrier integrity, thus allowing the microbial and its metabolite to translocate to the liver causing a high-end exposure of toxins and leading to hepatocyte injury [15,16]. The liver is a regenerative organ, and it tends to recover when proper measures are taken [17,18]. However, the continuous insult to the hepatocytes when the liver is at its most vulnerable state at subclinical pathological levels for instance lipid accumulation causes severe inflammation and critical physiological abnormalities in the liver [8]. This was further affirmed in retrospective clinical studies that correlated gut dysbiosis with the intestinal barrier and pathogenesis of NAFLD [13,19].

The gut harbors more than 100 trillion microbes in the human body (approximately 1.5 kg of total weight) making it one of the most diverse ecosystems [20]. The gut microbiota serves as key functional role in maintaining the homeostasis of the host’s metabolism, physiology, nutrition and immune-related functions, such as nutrient harvesting, energy regulation, vitamin synthesis, the fermentation of non-digestible fibers, bile acid metabolism and inflammatory response modulation [21,22]. The disruption of such host–microbe interaction and harmony leads to an array of chronic diseases, including alcoholic liver disease (ALD) and NAFLD [13] (Figure 1).

### 2.1. Gut Microbiota and Metabolic Mechanism

#### 2.1.1. Metaflammation and Lipotoxicity

Metabolic inflammation (metaflammation) is low-grade chronic inflammation that initiates a steady and long-term rise in low levels of inflammation identified by a small increase in immune system markers in the blood or tissue. Metaflammation is a hallmark of the pathological characteristics of a broad array of chronic conditions, including NAFLD. The microbes and their endotoxins translocate into systemic circulation as a consequence of small intestinal bacterial overgrowth (SIBO) and intestinal barrier disruption, which instigates low-grade inflammation. Unsurprisingly, metaflammation, which is considered sterile and noninfectious, has been identified as a causative factor in disease progression, and is involved in the transition from simple steatosis to NASH [23,24,25]. For instance, an animal study has shown that high-fat-diet-fed mice developed metabolic dysregulation and systemic inflammation. In this study, grouping was performed according to hyperglycemia and inflammation, with “responders” and “nonresponders” sorted according to normal blood glucose; the colonization of intestinal microbiota from responders to germ-free mice led to the development of hepatic steatosis, demonstrating that the microbial composition promotes the development of NAFLD [26].

Hepatocytes have an integrated response system to cope with the stress from pathogen invasion, nutrient fluctuations (especially lipids) or mitochondrial dysfunction [27]. Lipids play an important role in these homeostatic processes, but may also cause harmful metabolic consequences. Therefore, when the adaptive mechanism is overloaded with metabolic stress by prolonged lipid influx to the adipose tissue, lipids accumulate in hepatocytes and other ectopic sites. This harmful accumulation of lipids causes lipotoxicity, which initiates signaling that hinders immune regulation, thereby triggering immunometabolic dysregulation [24]. In addition, a clinical study has positively correlated the severity of NAFLD and NASH with adipose tissue inflammation, suggesting a requirement for the development of NASH [28].

#### 2.1.2. Insulin Resistance, Oxidative Stress and Mitochondrial Dysfunction

A sedentary lifestyle, unhealthy diet and obesity are the most important causes of functional defects in adipose tissue and the increased content of free fatty acids in circulation, resulting in the uptake of these free fatty acids by the liver, which causes hepatic steatosis. These features are linked with insulin resistance and are crucial pathophysiological factors that aggravate adipose tissue dysfunction, promoting lipogenesis in the liver and causing inflammation. This continuous stimulation of the inflammatory cascade further accelerates endoplasmic reticulum stress, the generation of reactive oxygen species (ROS) and sequential mitochondrial dysfunction in hepatocytes, which progresses mild liver injury to NASH and cirrhosis [9].

ROS are produced by cells during normal metabolic processes. During abnormal cellular function, ROS are not eliminated by cells, resulting in elevated levels of ROS in cells and disrupting the dynamic balance between antioxidants and prooxidants in oxidative stress [29]. In this vicious cycle, intestinal microbial components regulate disease at the circulating levels of the intestine, liver and whole body. With developments in the field of metagenomics, there is increasing evidence that the gut microbiota is correlated with insulin resistance and metabolic forms [30]. According to Angelini et al., the jejunum part of the small intestine regulates insulin sensitivity, and they found that patients who undergo metabolic surgery that involves bypassing the proximal small intestine have improved insulin resistance, thus increasing insulin sensitivity and lowering blood glucose levels [31].

Furthermore, a few studies have reported that some intestinal microbiotas stimulate the generation of ROS upon interaction with gut epithelia within the host cells [32]. The gut microbiota produces endogenous alcohol and its derived products, such as acetate and acetaldehyde, which promote morphological and functional alterations in the host cells, acting as an intestinal barrier and causing the leaky passage of endotoxins through the blood vessels. Upon reaching the liver, these endotoxins generate ROS by activating hepatic stellate cells and Kupffer cells [33,34]. Together with LPS, ROS promote increased TLR4 gene expression [35,36]. For instance, *Klebsiella pneumoniae*, which has been identified as a gut microbiota with high ethanol production, promotes liver disease in non-alcoholic individuals [37]. Along with the oxidative stress generated by endogenous alcohol, acetate promotes fatty acid synthesis, thus initiating fatty acid overload in hepatocytes, causing steatosis and subsequently triggering mitochondrial dysfunction and proinflammatory cytokine production [38]. Upon mitochondrial dysfunction, accumulated free fatty acids undergo partial metabolism through lipid peroxidation within the cells, resulting in other pro-oxidants, such as 4-hydroxy-2-nonenal and malondialdehyde, which have longer shelf lives than ROS and can easily spread to other parts of the body, amplifying intracellular and tissue damage and promoting NASH [39]. A preclinical study has suggested that gut microbes influence the redox state, potentially causing oxidative stress upon high-fat diet consumption. Treatment with antioxidants significantly decreases ROS and malondialdehyde levels in mice. Further analyzing the gut microbiota, Yi et al. found that *Escherichia coli* and *Enterococcus* are positively associated and *Lactobacilli* are negatively associated with the levels of ROS and malondialdehyde, suggesting that the microbiota is involved in the oxidative stress mechanism [40].

Oxidative balance is important for the proper function of the mitochondria. A recent report has suggested that oxidative balance regulates the metabolic pathways of antioxidants, especially the most important intracellular antioxidant, glutathione, in the host. Thus, a decrease in the intracellular concentration of glutathione has been implicated in oxidative stress [41]. During dysbiosis, the microbiota consumes the glutathione precursor, glycine, in the small intestine, which causes a deficiency in the production of glutathione, subsequently disrupting the redox balance between antioxidants and pro-oxidants and leading to oxidative stress. The bacterial genus *Clostridium sensu stricto* has been found to be one such bacterial strain that is markedly abundant in mice with NASH. However, replenishing glutathione by stimulating de novo glutathione synthesis with glycine-based tripeptide DT-109 treatment decreased *C. sensu stricto* abundance. Further DT-109 treatment also lowers lipogenesis and lipotoxicity, as well as enhancing FAO [42]. These results prompted additional studies to identify such bacterial strains. Moreover, glycine-based supplements with probiotics that increase the gut concentration of glutathione have also become one of the top interests for commercialization [43]. In addition, as intestinal mucosal cells in close contact with *Lactobacillus* genera, they cause further oxidation of proteins that serve as soluble redox mediator suppressors, such as glutathione and thioredoxin, leading to the induction of the nuclear factor erythroid 2-related factor 2 (Nrf2) antioxidant signaling pathway by upregulating the NRF2 transcription modulator [44]. *Lactobacillus rhamnosus* GG (LGG) has been found to augment the antioxidant pool intracellularly by producing 5-methoxyindoleacetic acid (5-MIAA), a small molecule, which then passes from the gut to liver and activates the Nrf2 transcription factor in hepatocytes. This augmentation is sufficient to prevent liver injury from drug-induced acetaminophen injury and ethanol toxicity. These results indicate a distinct signaling mechanism through which host–gut microbiota homeostasis impacts liver injury [45].

#### 2.1.3. De Novo Lipogenesis (DNL) Pathway

The metabolism of carbohydrates and lipids are closely linked to each other, especially in the liver. DNL is one such metabolic pathway that converts excess dietary carbohydrates into fatty acids that are stored as triacylglycerol and later utilized for energy production through β-oxidation [46]. Conventionally, during postprandial or abnormal conditions, DNL acts as the metabolic machinery that provides newly synthesized fatty acids for storage in the liver, or which releases lipoprotein into the bloodstream, contributing to disease progression [47,48,49]. DNL is strongly regulated by diverse lipogenic enzymes and transcription factors that act as potential safeguards to maintain the balance between fatty acid synthesis and β-oxidation. Hyperinsulinemia and the excess dietary intake of carbohydrates initiates DNL by supplying a large pool of substrates, which in turn stimulates the anabolic gene expression of lipogenic genes, such as ATP citrate lyase, acetyl-CoA carboxylase and fatty acid synthase, through the sterol response element-binding protein 1c (SREBP-1c) and carbohydrate response element-binding protein (ChREBP) transcription factors [49,50]. ChREBP performs a critical role in the production of acetyl coenzyme A (acetyl-CoA) from gut microbiota-derived acetate by activating acyl-CoA synthetase-2. Acetyl-CoA is carboxylated to malonyl-CoA by acetyl-CoA carboxylase, and malonyl-CoA undergoes a complex enzymatic process to produce long-chain fatty acids, which are used to make triacylglycerols. In the intestine, the dietary fructose is absorbed and metabolized to glucose. However, excess and unabsorbed fructose transitions to the colon where it is fermented to produce acetate by the intestinal microbiota, which serves as a precursor for acetyl CoA. Zhao et al. suggested a potential role for gut microbes in promoting fructose metabolism through alternative hepatic lipogenesis [51].

Low carbohydrate and high protein intake have been suggested to combat obesity and diabetes, as well as to promote weight loss in severely obese patients [52,53]. In contrast, recent studies have linked low carbohydrate/high protein intake to insulin resistance and type 2 diabetes mellitus with elevated branched-chain amino acid (BCAA) levels [54]. With insufficient BCAA levels, adjusting the level of other amino acids results in improved insulin sensitivity in mice [55]. In a recent clinical study, a high protein diet, particularly rich in glutamate, glutamine and one BCAA (leucine), was shown to contribute to DNL in healthy individuals, and an in vitro assessment has confirmed DNL via the activation of protein kinase B through the insulin signaling pathway [56]. In addition, another clinical study has demonstrated that *Bacteroides thetaiotaomicron*, which is a glutamate-fermenting bacterium, was significantly decreased and negatively correlated with serum glutamate in obese patients. Replenishing *Bacteroides thetaiotaomicron* through oral gavage in mice decreases serum glutamate and weight gain in high-fat-diet-fed mice, suggesting the possible involvement of the gut microbiota and glutamate metabolism [57].

### 2.2. Gut Microbiota and Immunologic Mechanism

#### 2.2.1. Lymphocytes and Macrophages

Because the gut is home to trillions of microorganisms, the intestinal mucosal immune system is specialized to combat gut-derived antigens, and its function is mostly local and independent of the systemic immune system, which undergoes important modifications after the colonization of the gut microbiota in the intestine [58]. The mucosal immune system provides a local immune microenvironment that can be defensive, tolerant or both. The liver, particularly rich with innate and adaptive immune cells, is the main immunological organ, with high exposure to circulating endotoxins and antigens from the gut microbiota [59]. The intestinal mucosa acts as a first line of the biophysical barrier in the containment of undesired luminal substances within the luminal wall, while conserving the ability to absorb essential nutrients [60]. This barrier is further reinforced by the presence of various immune cells that can be grouped as intraepithelial and lamina propria (LP) cells, which further contribute to the intestinal barrier. The intraepithelial cells include intraepithelial lymphocytes (IELs), constituting T cell receptor-positive (αβTCR+ and γδTCR+) and T cell receptor-negative (TCR-) IELs, as well as intraepithelial mononuclear phagocytes. These TCRs are further categorized under the following conditions: (1) conventional IELs that co-express the αβTCR and CD4 or CD8αβ receptors and which are activated upon the recognition of foreign antigens by effector cells; and (2) unconventional IELs that express TCR+ receptors (αβTCR or γδTCR) and which are activated upon self-antigen stimulation. Conventional IEL subsets represent a comparatively smaller population of total intestinal IELs [61,62]. Upon foreign antigen recognition, the accumulation and activation of conventional IELs occur in the intestine, and this accumulation of conventional IELs is distinctly reduced in germ-free mice, suggesting that the antigen is likely to be derived from the gut microbiota or dietary intake [62,63]. Additionally, in germ-free mice, a significant decrease in the levels of antimicrobial peptides and secretory immunoglobulin A, which provide protective intestinal mucosal immunity, has been observed; however, this level is replenished upon colonization with commensal microbiota, suggesting crucial host immune–microbiota interactions [58,64]. Commensal microbiota not only participate in the host immune response, but also shape them indirectly. For instance, Cervantes-Barragan et al. demonstrated the effect of *Lactobacillus reuteri* (*L. reuteri*) on the differentiation of conventional TCR+ cells IELs. *L. reuteri* utilizes dietary tryptophan and metabolizes it into indole derivatives. This indole derivative activates a ligand-activated transcription factor, the aryl-hydrocarbon (AhR) receptor, which regulates intestinal immunity and inflammation [65]. The activation of AhR in CD4+ T cells facilitates its differentiation into TCRαβ + CD4 + CD8αα + IELs, implying that *L. reuteri*, with a tryptophan diet, can reprogram and expand CD4+ T cells into immunoregulatory cells in the LP and epithelium. This phenomenon was observed in mice housed in different facilities purchased from two different vendors [66]. Furthermore, it has been documented that some gut bacteria, such as the genus *Bifidobacterium*, affect Treg cell development, while some segmented filamentous bacteria promote T-helper cell-17 development [67].

Under normal circumstances, these microbes and their derived products are confined to the luminal wall by the mucosal immune system. However, in the presence of gut dysbiosis, intestinal inflammation occurs and the intestinal barrier ruptures, leading to the translocation of bacteria, whereby their endotoxins reach to the liver, where they are eliminated by the local action of liver-resident macrophage–Kupffer cells and patrolling blood monocytes. Thus, the liver acts as a second line of defense in the elimination of bacteria, and their derived products from circulation compromise intestinal immunity [68]. During liver dysfunction, this second line of defense is compromised, resulting in the failure of bacterial clearance and, thus, the leakage of bacteria into systemic circulation. This increased systemic exposure to bacteria initiates the priming of the systemic immune response. The resultant inflammatory activation of Kupffer cells and hepatic stellate cells recruits additional innate and adaptive immune cells to the liver, including natural killer (NK) cells, neutrophils, cytotoxic T cells, monocytes and NK = T cells. This consistent systemic immune response has been implicated in NAFLD [59]. Monocytes in the liver differentiate into CD11b+F4/80+ proinflammatory macrophages (M1-type), which secrete proinflammatory cytokines and have phagocytic activity [69]. M1-type macrophages are mostly stimulated by TLR ligands, such as LPS and interferon-gamma [70]. Gut-derived endotoxins, SFAs and lipid metabolites are the major conducive factors for macrophage activation in NAFLD [71]. An in vitro experiment has demonstrated that isolated Kupffer cells from choline-deficient and amino acid-defined (CDAA) diet-fed mice show elevated production of TNF-a upon LPS stimulation compared to cells from normal control mice [72]. Correspondingly, chemotactic analysis of isolated lipid-laden Kupffer cells from mice fed a high-fat diet has shown a higher percentage of migrated lymphocytes after LPS stimulation compared to cells isolated from normal control mice. This study demonstrated that chronic liver inflammation was due to macrophages and the hyperresponsiveness of migrated lymphocytes in the liver, and provided a possible link between lipotoxicity and macrophage activation [73].

Similarly, leptin can also trigger the activation of Kupffer cells by a peroxynitrite-dependent mechanism that can prompt inducible nitric oxide synthases to produce nitric oxide. This nitric oxide further reacts to produce a potent biological oxidant, peroxynitrite, which skews activated Kupffer cells to the M1 macrophages [74]. In contrast, M2 macrophages are observed in the reformative stage of NASH; M2 macrophages have an immunosuppressive function, but they are profibrogenic and secrete high levels of interleukin-13 and transforming growth factor-β1 (TGF- β1), subsequently influencing the progression of fibrosis [75,76]. A previous study has shown that M2 macrophages alleviate NAFLD and ALD by initiating the apoptosis of M1 macrophages via an arginase-dependent pathway [77], thus limiting hepatocyte injury during chronic inflammation. Hence, dynamic changes in the macrophage phenotype are associated with the pathophysiology of NAFLD [77].

The intrahepatic accumulation of cytotoxic CD8+ T cells has been linked with decreased insulin sensitivity and gluconeogenesis in high-fat-diet-induced obese mice, and it is mainly initiated by the type I interferon (IFN) response [78]. Unmethylated cytosine phosphate guanosine microbial motifs, which are abundant in the prokaryotic DNA found in the gut microbiota [79], increase the fatty acid oxidation and oxidative phosphorylation facilitated by type I IFN signaling via TLR-9 in plasmacytoid dendritic cells [80]. Conversely, in the liver, the induction of type I IFN signaling via the activation of TLR-9 through interferon regulatory factor-7 shows an anti-inflammatory response by attenuating liver injury [81]. These studies suggested that different responses from type I IFN signaling may be associated with the functional specificity of the cell type and the surrounding microenvironment of the host immune response.

In a clinical study exhibiting gut microbial impact on peripheral immune response in NAFLD and NAFLD-HCC patients, Behary et al. provided evidence that metabolites, such as short-chain fatty acids (SCFAs), can elicit an immunoregulatory response [82]. The compositional and functional shifts in NAFLD patients tended to drive HCC through an immunosuppressive response, as regulatory T cells (Treg) were significantly higher and cytotoxic CD8+ T cells were significantly lower than those in NAFLD-cirrhosis patients. This observation was further confirmed through the ex vivo stimulation of bacterial extracts from NAFLD-HCC patients in the peripheral mononuclear cells isolated from healthy controls, which showed that cytotoxic cell expansion was suppressed; however, the extract induced greater CD4+CD25+ Treg cell expansion than the bacterial extracts prepared from NAFLD-cirrhosis, which had a negligible effect on isolated cells [82]. *Bacteroides*, *Veillonellaceae,* and *Ruminococcus* are correlated with proinflammatory markers, inflammation and fibrosis markers in NAFLD patients [83,84], *Bacteroides xylanisolvens* and *Veillonella parvula* have shown a significant positive correlation with Treg cells, and *Ruminococcus gnavus* has a positive correlation with Treg cells but a significant negative correlation with cytotoxic CD8+ T cells, suggesting a gut microbiota immunomodulatory response in NAFLD-HCC patients.

#### 2.2.2. Chemokines

Immune cell influx to the liver is largely regulated by chemokines [85]. Chemokines are a family of chemoattractant cytokines or small secreted proteins that perform an important function in cell movement through blood vessels from blood into tissue at the site of inflammation, and vice versa. This induction of cell migration or movement occurs in response to the chemokine gradient by a process called chemotaxis or chemotactic migration [86]. These chemokines also mediate the production and secretion of inflammatory mediators and promote lymphoid tissue maturation [87]. Kupffer cells secrete chemokines, such as C-C motif ligand (CCL)-2, also referred to as monocyte chemoattractant protein-1 (MCP-1), which recruits monocytes to the liver and induces their differentiation into monocyte-derived macrophages. CCL2, along with its cognate receptor C-chemokine receptor (CCR)-2, forms a CCL2-CCR2 complex leading to the recruitment and accumulation of macrophages and, subsequently, insulin resistance and hepatic steatosis in obese patients [88,89]. Moreover, in a preclinical study of a high-fat diet model, Morinaga et al. found two distinct types of morphology in Kupffer cells and recruited macrophages; the CCL2/MCP1 ligand was highly upregulated in Kupffer cells, whereas CCR2 expression was five times increased in recruited macrophages [90]. This study was further supported by clinical findings that CCR2 is significantly expressed in recruited macrophages and not in Kupffer cells in more severe NAFLD patients [91]. These recruited macrophages establish a relationship in the enhancement of hepatic insulin resistance and obesity-induced inflammation, as these distinct features of two different types of macrophages have been observed in lean mice [90]. Thus, the reduction of the recruitment of monocytes to the liver may lessen the burden of NASH progression in NAFLD.

Recently, a previous study has revealed the mechanism utilized by the gut microbiota in the progression of tumorigenesis. Gut microbes use LPS to trigger the accumulation of monocyte-derived macrophages via the upregulation of epithelial CCL2 in the gut, which is abolished by antibiotics. The microbiota-derived signals for monocytes to differentiate into monocyte-derived macrophages may be essential for the upregulation of CCL2 and the inflammatory environment. Thus, the gut microbiota favors the production of chemokines and acts as a major source for the upregulation of CCL2 in macrophages through LPS/TLR-4 pathways [92]. Consistently, systematic reviews and network meta-analyses have revealed that the significantly increased expression of CCL2 and C-X-C motif ligand 8 (CXCL8, which is secreted by macrophages and functions in the recruitment of neutrophils and monocytes [93]) may be associated with NASH and fatty liver, compared to normal controls. CXC chemokines contribute to acute inflammation, while CC chemokines facilitate chronic inflammation. Interestingly, evaluating chemokines at an early stage may introduce different prospects to target NAFLD through pharmacological interventions [94]. The regulation of gut microbiota through chemokines has been further elucidated. A previous study has found that the gut microbiota control NK-T cell infiltration into the liver via CXCL16 expression in liver sinusoidal epithelial cells through primary bile acids. Given that CXCR6 is the only cognate receptor for CXCL16, the infiltration of CXCR+NK-T cells in the liver promotes anti-tumor activity in a liver metastasis model. In contrast, secondary bile acid reduces the expression of CXCL16 and, subsequently, the progression of tumors. Abolishing Gram-positive bacteria, which are the major source for the conversion of primary bile acids (BAs) to secondary BAs, through antibiotic treatment promotes hepatic NK-T accumulation, resulting in tumor reduction. *Clostridium scindens* has been found to enhance the conversion of primary BAs to secondary BAs, impair the primary/secondary BA ratio, and to consequently reduce NK-T cell accumulation in the liver. A dysregulated primary to secondary BA ratio has been observed in liver cancer patients, in which chenodeoxycholic acid (primary BA) levels are positively correlated with CXCL16 expression, and glycolithocholate (secondary BA) levels are negatively correlated with CXCL16 expression [95]. These studies suggest that the gut microbiota shapes immunity through the recruitment of immune cells or metabolic processes.

### 2.3. Gut Microbiota and Its Metabolites

#### 2.3.1. Fermentable Dietary Nutrients

For bacterial–host communication, the gut microbiota produces many small molecules through the fermentation of indigestible polysaccharides. SCFAs are the most commonly studied and most abundant metabolites that are produced by bacterial metabolism [96]. Acetate, butyrate and propionate are metabolites that comprise more than 95% of the total pool of SCFAs, demonstrating diverse functional roles [97,98]. In addition to providing energy to the gut epithelium, SCFAs have other bioactive functions, such as glucose metabolism, lipid metabolism and immune regulation, in maintaining microbiota homeostasis [99]. Additionally, some branched-chain fatty acids, such as isovalerate, 2-methylbutyrate, isobutyrate, succinate and lactate (propionate intermediates), are also produced in low proportions and have biological effects [100]. Acetate, the most abundant SCFA, is produced from pyruvate mainly by the enriched species within the phylum Bacteroidetes [97,101]. Propionate, another major SCFA, is produced by the propionogenic microbial association of *Ruminococcus obeum*, *Lactobacillus plantarum*, *Akkermansia muciniphila*, *Bacteroides thetaiotaomicron*, *Bacteroides vulgatus*, *Coprococcus catus*, *Veillonella parvula* and Clostridial cluster IX from sucrose via the succinate pathway and lactate via the acrylate pathway [97,101,102] Another major SCFA, butyrate, is produced by obligate anaerobes from Clostridial consortia, of which *Faecalibacterium prausnitzii* (*F. prausnitzii*) are the most abundant groups [101]. Butyrate is mainly produced via the succinate, lysine, glutamate and acetyl-CoA pathways. Some bacteria, such as *Coprococcus catus*, *Eubacterium rectale*, and *Roseburia* spp., use acetates, while other bacteria, such as *Anaerostipes* spp., *Eubacterium hallii*, and *F. prausnitzii*, use both acetate and lactate to produce butyrate [97].

Some findings have suggested beneficial effects of SCFAs for lowering lipogenesis in adipocytes and improving insulin sensitivity in obese mice [103,104]. Contradicting these studies, other studies have suggested that, during metabolic dysfunction, SCFA concentrations are higher in obese patients than in lean individuals, demonstrating a correlation of higher concentrations of SCFA with a high body mass index [105]. Supporting these clinical findings, obese mice have elevated fecal SCFAs [106]. NAFLD is highly associated with obesity and insulin resistance [7], and fecal SCFA concentration is consistently higher in NASH patients [107]. Another finding has suggested that acetate serves as a precursor to initiate DNL in hepatocytes by the action of acetyl-CoA synthetase short-chain family member 2 (ACSS2) to generate a lipogenic acetyl-CoA pool. Abolishing the microbiome or suppressing ACSS2 significantly suppresses hepatic lipogenesis in mice [51]. In contrast to its function in the liver, acetate has also been shown, via the gut–brain axis, to reduce appetite by increasing γ-aminobutyric acid action, consequently reducing weight, indicating a potential therapy for obesity through central appetite regulation [108]. The precise roles of SCFAs have been established; however, the roles of SCFAs are controversial in NAFLD. Wang et al. reported reduced SCFAs in fecal microbiota in nonobese NAFLD patients compared to healthy nonobese individuals [109]. In agreement, another study has reported that SCFAs were significantly reduced in fecal microbial samples in NAFLD patients, but this reduction is restored in a Western diet model upon supplementation with *Lactobacillus lactis* and *Pediococcus pentosaceus* [110]. Supplementation with sodium acetate [111] and sodium butyrate [112] significantly reduces hepatic steatosis and inflammation, thereby alleviating hepatic injury in mice, and targeted propionic ester in obese individuals reduces hepatic lipid accumulation [113], indicating a possible mutual establishment between the gut microbiota and the liver. In light of these findings, Endo et al. reported that a butyrate-producing probiotic, MIYAIRI-588, which contains *Clostridium butyricum*, reduces NAFLD in CDAA diet-fed rats by increasing the activation of adenosine 5′-monophosphate-activated protein kinase (AMPK) in the liver, leading to increased β-oxidation and, subsequently, decreased hepatic lipid accumulation. Additionally, probiotic treatment suppresses oxidative stress via the significant upregulation of Nrf2, which stimulates antioxidative enzymes in hepatocytes, thereby ameliorating the progression of NAFLD [114]. A prospective study of biopsy-proven NAFLD patients has reported that the inosine and hypoxanthine plasma metabolites are elevated in moderate NAFLD patients, while malate, α-ketoglutarate, succinate, glutamine, lactate, fumarate, serine, α-ketobutyrate and glutamate (associated with the carbon metabolism pathway and detoxification) are elevated in advanced fibrosis in NAFLD, providing novel noninvasive biomarkers for the early detection of an advanced fibrosis state from a fecal microbiome-derived metagenomic signature in NAFLD patients [115].

While SCFAs are associated with different metabolic pathways, their functions have been implied in signaling mechanisms through G-protein-coupled receptors (GPCRs), mainly GPCR-41 (free fatty acid receptor 2 (FFAR3)) and GPCR-43 (FFAR2), which are expressed on intestinal, liver, adipose and other tissues [116,117]. FFAR2 is a known receptor for acetate in hepatocytes. The enrichment of *Bacteroides acidifaciens* and *Blautia producta* via prebiotic inulin consumption synergistically elevates the concentration of acetate in mice fed a high-fat/high-fructose/high-cholesterol diet, which remarkedly suppresses hepatic steatosis and fibrosis in mice via CCL2 expression. As a result, the recruitment of infiltrating immune cells, such as monocyte-derived macrophages and CD8+ T cells, to the liver is reduced. Additionally, fibrotic gene expression of the α-smooth muscle actin 2 (αSMA) and TGF-β gene was significantly decreased by inulin. The deletion of liver-specific FFAR2 worsens insulin resistance, causing inflammation, hypercholesterolemia and liver fibrosis, suggesting a protective effect of GPCR activity mediated by acetate and FFAR2 [118]. An in vitro study has identified butyrate as a ligand of AhR in human intestinal epithelial cells, thus functioning in maintaining intestinal barrier integrity [119]. In addition, SCFAs, in particular butyrate and acetate, also perform immunomodulatory functions by modulating peripheral immune response in NAFLD [82,118].

#### 2.3.2. Amino Acid Metabolism and Its Byproducts

Despite effective protein assimilation in the small intestine from dietary protein, 5–10% of proteins are not absorbed and transported to the colon, where the metabolism of amino acids (AAs) is accomplished by the gut microbiota. If the degree of bacterial fermentation of proteins is higher than that of carbohydrates, fewer SCFAs are produced, which results in a high colonic pH, thereby leading to altered composition and functions of the gut microbiota [120]. Tryptophan, an essential amino acid, has recently attracted attention in the field of liver diseases. The metabolism of dietary tryptophan by the gut microbiota produces various signaling molecules, such as kynurenines, tryptamine, serotonin, melatonin and indole compounds, which may participate in host–microbial communication through the gut–brain–liver axis [100,121,122]. Recent studies have suggested that indole and its compounds, such as indole-3-acetic acid (I3AA), indole-3-propionic acid (I3PA) and tryptamine, have profound protective effects in NAFLD. In particular, I3AA is mainly synthesized by *Bacteroides* species and *Clostridium bartletti* [123]. At the cellular level, treatment with I3AA and tyramine alleviates the production of proinflammatory cytokines and fatty acids, and it inhibits the movement of cells toward MCP-1 in LPS-stimulated macrophages. The effect of I3AA is more potent than that of tyramine in macrophages. Additionally, I3AA treatment also lowers the lipogenic gene expression of FAS and SREBP-1, thereby reducing lipid accumulation in hepatocytes and suppressing the inflammatory response. It has been observed that these effects are AhR-dependent [124]. Indole compounds serve as one of the broad range of ligands for the activation of AhR and lipid, glucose and cholesterol homeostasis in the gut and liver [125,126]. *Clostridium* and *Peptostreptococcus* species synthesize I3PA [123]. The I3PA treatment of LX-2 human hepatic stellate cells in vitro significantly reduces the mRNA gene expression of type I collagen (*COL1A2*), integrin subunit alpha 3 (*ITGA3*) and *αSMA*, which are required for the activation, cell migration and cell adhesion of LX-2 cells upon TGF-β1 stimulation, implying that I3PA has a potential function as an antifibrotic agent in NASH [127]. Another indole compound, indole-acrylic acid (IAA), which is synthesized by *Peptostreptococcus* species, mitigates the proinflammatory response and enhances tight junction protein in the gut, thereby reducing leakiness following the maintenance of intestinal barrier integrity [128]. Additionally, glycine-based DT-109 treatment enhances FAO and improves steatohepatitis by stimulating de novo glutathione synthesis in mice [42]. A glutathione decrease is associated with disease severity in NAFLD patients [129]. These studies provide insight into mitigating disease conditions in the liver; however, the signaling cascade involved in the alleviation of such conditions in the liver through tryptophan metabolites and other AAs needs to be further elucidated for better outcomes when considering therapeutic treatment in NAFLD.

#### 2.3.3. Bile Acids as Gut Microbial Messengers

Primary BAs are synthesized in the liver from cholesterol by the rate-limiting enzyme cholesterol 7 alpha-hydroxylase (CYP7A1), and they are stored in the gallbladder and released into the proximal duodenum through the bile duct to aid digestion. Primary BAs are then converted to secondary BAs by resident bacteria through deconjugation and dehydroxylation. Up to 95% of the primary BAs are reabsorbed through the distal ileum and transported back to the liver via hepatic portal system–enterohepatic circulation [130]. Some BAs remain in systemic circulation and exert signaling mechanisms through the activation of specific BA receptors, including members of the nuclear receptor superfamily (farnesoid X receptor (FXR), vitamin D receptor and pregnane X receptor) and members of the GPCR superfamily (Takeda G protein-coupled receptor 5 (TGR5) and sphingosine-1-phosphate receptor 2) [131,132]. The activation of FXR serves as a feedback system for regulating BA production. In intestinal epithelial cells, FXR binds to BAs and initiates the transcription process of fibroblast growth factor, which enters the liver via enterohepatic circulation and suppresses the synthesis of BAs in hepatocytes through the inhibition of the CYP7A1 enzyme. This activation of FXR reduces mucosal inflammation and modulates the microbiota [133]. Furthermore, BA-induced TGR5 activation enhances glucagon peptide 1 release, thereby promoting insulin release, which lowers insulin resistance, decreases inflammation and improves liver functions [134].

Numerous clinical trials and preclinical experiments have confirmed that dysbiosis causes altered BAs, which have crucial roles in the progression of NAFLD [132]. Additionally, cross-sectional studies in biopsy-proven NAFL and NASH patients have analyzed fecal and plasma BA concentrations, which are significantly altered in NASH [135,136]. Total fecal primary BAs (cholic acid (CA) and chenodeoxycholic acid (CDCA)) and the ratio of primary BAs to secondary BAs are elevated in NASH patients. The abundance of *Clostridium leptum,* which performs 7α-dehydroxylation and deconjugation in the colon to convert primary BAs into secondary BAs, [137] is significantly decreased in NASH patients and is inversely correlated with CA and CDCA [135]. In plasma, modified circulating BAs are associated with NAFLD and positively correlated with the histology of NASH, suggesting that BAs may contribute to the progression of NAFL to NASH [136]. However, Lee et al. observed distinct features of microbial metabolites between obese NAFLD and nonobese NAFLD patients. Total Bas (conjugated and unconjugated), including CA, CDCA, glycochenodeoxycholic acid, ursodeoxycholic acid (UDCA) and glycoursodeoxycholic acid, are significantly elevated in nonobese NAFLD patients and are positively associated with the worsening of fibrosis severity. Secondary Bas, namely, lithocholic acid and deoxycholic acid, are elevated in obese NAFLD patients with significant fibrosis. Distinct species, namely, *Ruminococcus bromii, F. prausnitzii* and *Roseburia intestinalis*, are inversely related with primary BA concentration and fibrosis severity; conversely, *Megamonas* spp. exhibits a positive association with UDCA and progressive fibrosis severity in nonobese NAFLD patients [84].

#### 2.3.4. Other Bacterial Metabolites

Choline from dietary sources, such as eggs, red meat and dairy products, is an essential nutrient for the host and is metabolized by microbiota to produce trimethylamine (TMA). In the liver, TMA is oxidized to produce trimethylamine N-oxide (TMAO) by hepatic flavin monooxygenase [138]. The following bacterial species have been isolated and found to produce TMA in vitro: *Clostridium* spp., *Anaerococcus hydrogenalis*, *Escherichia fergusonii*, *Proteus penneri*, *Providencia rettgeri*, and *Edwardsiella tarda* [139]. To provide evidence that TMA is a bacterial metabolic product and is produced by bacteria, a conventional diet has been supplemented with choline in germ-free mice. Compared to normal mice, choline-supplemented mice do not show an increase in the concentration of TMA (which is the precursor of TMAO), indicating that TMAO can only be synthesized by bacterial metabolism [140]. Studies have revealed that a higher circulating TMAO concentration is positively associated with fatty liver index and all-cause mortality in NAFLD [141]. Given the important involvement of TMAO in disease progression, Aragonès et al. have suggested that TMAO can be used as a “liquid biopsy” in the prognosis of NASH [142]. Preclinical studies agree with the clinical findings that TMAO increases insulin resistance and hepatic lipid accumulation and compromises liver functions [143]. However, contrary to this, Zhao et al. demonstrated that oral TMAO alleviates high-fat-diet-induced steatohepatitis in mice by inhibiting intestinal cholesterol absorption and reducing cell death upon cholesterol overload [144]. More studies are required to comprehend the mechanism and the influence of TMAO in NAFLD.

Fermenting bacteria produce endogenous ethanol from dietary carbohydrates, and a higher abundance of fermenting bacteria under dysbiosis conditions produces more endogenous ethanol. *Klebsiella pneumonia* is one of the species that can produce ethanol similar to other ethanol-producing bacteria, such as *Bifidobacterium adolescentis*, *Clostridium thermocellum*, *Escherichia* and *Bacteroides fragilis* [145]. From the fecal microbiome, species *K. pneumonia* from the *Proteobacteria* phylum has been found to be the cause of fatty liver in patients with NAFLD and NASH. Additionally, before fecal microbial transplantation (FMT), such bacteria need to be diminished or prevented to clinically improve steatosis [37].

## 3. Therapeutic Approaches

### 3.1. Pharmacological Intervention

Given the multifactorial and wide array of complex pathophysiology of NAFLD, its diagnosis has been difficult. Hence, the successful treatment of patients with NAFLD at different stages is challenging. Therefore, varied individual therapies targeting NAFLD at a particular stage are recommended at the individual level [146,147]. According to the EASL-EASD-EASO Clinical Practice Guidelines, pharmacological drug therapy is suggested for progressive NASH (≥fibrosis stage-2). Additionally, patients with early-stage NASH with metabolic syndrome, diabetes mellitus or increased liver function should be enrolled for pharmacological drug therapy, as they have a high risk for disease progression [147,148]. Pathophysiological drug therapies for NAFLD are under development, but response rates are the reason for the lack of approved drug treatments. These drug therapies appear to have modest effects, primarily for the treatment of fibrosis. Despite intensive clinical studies, there are currently no Food and Drug Administration-approved drugs for NASH, and no particular therapy can be suggested. The currently prescribed drugs for NASH are being used off-label worldwide [146,147,148].

### 3.2. Microbiota-Based Intervention

After decades of intensive pharmacological interventions trying to identify targeted drugs for the treatment of NAFLD, scientists across the world are seeking different approaches for the treatment of NAFLD. Such treatments include the use of probiotics as microbial interventions. The influence of gut microbiota has led to numerous preclinical and clinical studies for the effective prevention and treatment of NAFLD, NASH and NAFLD-HCC (Table 1 and Table 2) [149,150]. Nevertheless, various probiotics, such as *Lactobacillus*, *Bifidobacterium* and *Pediococcus*, have demonstrated beneficial effects in the abrogation of NAFLD in preclinical models. The administration of probiotics in rodents abolishes NAFLD by restoring microbial homeostasis in the gut, which reduces lipogenesis, subsequently lowering inflammation in the liver [110,151,152]. Over the past decade, numerous clinical trials have been conducted to optimize and translate the beneficial effect of probiotics in diseased conditions in humans; however, translating microbiota-based therapy is still under development. In a recent systematic review and meta-data analysis that summarized microbiota-based targeted therapy (9 probiotics and 12 synbiotics) in NAFLD patients from 21 randomized controlled trials, Sharpton et al. found that probiotics or synbiotics are positively associated with improvements in liver function enzymes, hepatic steatosis and liver stiffness measurements (which reflect NASH conditions). Given the heterogeneity of the population, varying liver disease phenotypes and varying durations of probiotic or symbiotic interventions, probiotics and synbiotics may be able to produce liver-specific beneficial outcomes [153]. However, microbiota-based targeted therapy is still limited to preliminary clinical studies as a result of their relatively small scale and heterogeneous population, random dosing and diverse indicators/endpoints. The latest evidence suggests that transplanting microbiota based on the individual gut environment, in which the indigenous microbiota show greater environmental validity and sustainability in that individual, offers prospects for personalized microbiome reconstitution [154].

Other available strategies that are focused on altering gut microbial compositions for beneficial effects in NAFLD are prebiotics, pre- and probiotic combinations (synbiotics), antibiotics and FMT. Prebiotics are indigestible fermentable dietary fibers that are selectively utilized by the gut microbiota to confer host–microbial benefits. Short- and long-chain β-fructans (inulin and fructooligosaccharides (FOS)), galactooligosaccharides (GOS) and lactulose are common prebiotics. Prebiotics are the most well studied in chronic liver disease. In this disease, the administration of lactulose enhances life expectancy by reducing the possibility of hepatic encephalopathy (HE) recurrence and HE-related hospitalization by managing symptomatic hyperammonemia in cirrhosis with HE patients [155]. Interestingly, Sarangi et al. recently revealed that there are no changes in the composition of the gut microbiome after lactulose administration in cirrhosis patients, suggesting that the effect of lactulose in HE may not be related to the compositional changes in the gut microbiome [156]. In contrast, the antibiotic, rifaximin, which is also used in HE management, has minimal compositional changes with improved endotoxemia and cognitive functions [157]. Interestingly, a meta-data analysis of RCTs for synbiotic supplementation in NAFLD patients has shown beneficial effects on lipid profiles, liver function enzymes and inflammatory parameters [158]. Regardless of the promising aspects of synbiotics, a recent study has revealed that synbiotics (FOS with *Bifidobacterium animalis* subspecies *lactis* BB-12) only modulate the gut microbiome without reducing steatosis or fibrosis in patients with NAFLD [159]. In contrast, another clinical study demonstrated that symbiotic supplementation in lean NAFLD patients significantly reduced hepatic steatosis and fibrosis compared with placebo treatment. Other inflammatory markers and lipid profiles were also significantly reduced in comparison to the placebo group [160]. These two studies suggest that significant metabolic dysfunctional factors may hinder the resolution of hepatic injury markers, and further studies need to be performed to determine if the metabolic factors have any role in the primary outcome of the study. Furthermore, modifying the microbial environment by FMT has also been attempted for NAFLD treatment. A clinical report by Vrieze et al. demonstrated that six weeks after receiving allogenic FMT from lean donors, enhanced insulin sensitivity in participants with metabolic syndrome was accompanied increased levels of butyrate-producing bacteria, *Roseburia intestinalis* and *Eubacterium hallii* [161]. Additionally, *Roseburia intestinalis* was found to be negatively correlated with fibrosis and disease severity in non-obese NAFLD [84]. In addition, a recent double-blinded, randomized clinical study also showed that allogenic FMT from healthy donors significantly decreased the serum GGT levels and improved necrosis in liver histology in the patients with NAFLD compared to the autologous FMT. Additionally, significant changes were observed in the hepatic gene expression, which is responsible for the maintenance of endothelial integrity after allogenic FMT, suggesting that clinicians broaden the undiscovered horizon for therapeutic purposes [162]. A summary of functional studies of the microbiota in the prevention and progression of NAFLD in animals and humans is listed in Table 1 and Table 2, respectively.

**Table 1 biomedicines-10-00550-t001:** Microbiota-based preclinical studies in non-alcoholic fatty liver disease.

Animal Study Model	Intervention	Co-Intervention	Effect on Gut	Effect on Liver Function	Metabolites	Ref.
HFD-NAFLD	Astragalus polysaccharides (prebiotic)	-	*Desulfovibrio*↑, *Parabacteroides*↑, *Acetatifactor*↑, *Alistipes*↑, F/B ratio↓	TG↓, ALT↓, hepatic steatosis↓, hepatic inflammation↓, fatty acid oxidation↑	Acetate↑	[163]
HFD-NAFLD	*Desulfovibrio vulgaris*	-	*Desulfovibrio vulgaris*↑,	TG↓, ALT↓, hepatic steatosis↓, fatty acid oxidation↑		[163]
HFrD-NAFLD	*Lactobacillus fermentum* CECT5716	FOS	F/B ratio↓, intestinal barrier integrity↑, *Lactobacilli*↑	insulin resistance↓, hepatic steatosis↓, hepatic inflammation↓	SCAs↓	[164]
HFD-NAFLD	*Lactobacillus acidophilus, Lactobacillus fermentum, Lactobacillus plantarum*	-	F/B ratio↓, *Lactobacillus*↑	TC↓, TG↓	-	[151]
HFD-NAFLD	*Bifidobacterium bifidum* V *Lactobacillus plantarum* X	*Salvia miltiorrhiza* polysaccharide	Fecal TC↑, *Cyanobacteria*↓, F/B ratio↓	TC↓, TG↓, LPS↓, hepatic steatosis↓, insulin resistance↓, hepatic inflammation↓	SCAs↑	[165]
HFD-NAFLD	Kefir (probiotic beverage)	-	*Lactobacillus/Lactococcus↑*, *Bacteroides* fragilis↓, *Clostridiaceae*↓, F/B ratio↓	TC↓, fatty acid oxidation↑, hepatic inflammation and oxidative stress↓	-	[166]
HFD-NAFLD	*Bacillus* mixture VSL#3	-	Intestinal barrier integrity↑, *Bacteroidetes*↑	Hepatic steatosis↓, insulin resistance↓, hepatic inflammation↓, fatty acid oxidation↑	Acetate↓	[167]
HFD-NAFLD	*Lactobacillus rhamnosus* GG	-	*Desulfovibrionaceae↑*, *Lactobacillaceae↑*	LPS↓, hepatic steatosis↓	FAs↓	[168]
HFD-NAFLD	*Lactobacillus lactis*, *Pediococcus pentosaceus*	-	Intestinal barrier integrity↑, F/B ratio↓, *Clostridium*_g21↓	TG↓, AST↓, TBil↓, TC↓, LPS↓, hepatic steatosis↓, hepatic inflammation↓, fatty acid oxidation↑	Indole compounds↑ Acetate↑ Butyrate↑ Propionate↑ Primary BAs↑	[110]
HFD/HFrD-NAFLD	*Lactobacillus plantarum* K2 and K6	-	*Bacteroides*↑	ALT↓, AST↓, ALP↓, TC↓, TG↓, MDA↓, hepatic steatosis↓, fatty acid oxidation↑, oxidative stress↓	-	[169]
HSuD/HFD-NASH	*Lactobacilli* (9 species), *Bifidobacteria* (4 species), *Streptococcus salivarius* subsp (*Thermophilus*)	Inulin	*Bacteroides*↑	ALT↓, AST↓, GGT↓, ALP↓, TBil↓, TC↓, TG↓, hepatic steatosis↓, fibrosis↓, hepatic inflammation↓	-	[170]
HFD-NASH	*Lactobacillus reuteri*	Metformin	*Bacteroidetes*↓, *Firmicutes*↑	ALT↓, AST↓, TC↓, TG↓, LPS↓, insulin resistance↓, oxidative stress↓, hepatic steatosis↓, fibrosis↓, hepatic inflammation↓	Acetate↓ Butyrate↑ Propionate↑	[171]
HFD-NAFLD	Polylactose	-	*Bacteroides*↑, *Lactobacillus*↑, *Akkermansia muciniphila*↑ F/B ratio↓,	TC↓, TG↓, insulin resistance↓, hepatic steatosis↓, hepatic inflammation↓	Acetate↑ Propionate↑	[172]
HFD-DSS-NAFLD	*Schizophyllum* *commune*-derived β-glucan	Probiotic mix (8 species)	*Lactobacillus*↑, *Bifidobacterium*↑, *Akkermansia*↑	ALP↓, hepatic steatosis↓, hepatic inflammation↓	Butyrate↑	[173]
CDAA-NASH	MIYAIRI 588	Losartan	Intestinal barrier integrity↑	Hepatic steatosis↓, hepatic inflammation↓, fibrosis↓, early HCC↓	-	[174]
HCholD	*Lactobacillus paracasei*, *Lactobacillus rhamnosus*, *Lactobacillus acidophilus*, *Bifidobacterium lactis*	FOS	-	TC↓, hepatic steatosis↓, hepatic inflammation↓	-	[175]
MSG one dose (s.c.)	FOS	-	*Clostridium* cluster XI↑ *Prevotella*↓	TC↓, ALT↓, LPS↓, insulin resistance↓, hepatic steatosis↓, hepatic inflammation↓	Acetate↑ Butyrate↑ Propionate↑	[176]

Abbreviations: ↑ indicates an increase in the condition or level; ↓ indicates a decrease in the condition or level; ALT, alanine transaminase; AST, aspartate aminotransferase; HFD, high-fat diet; F/B ratio, *Firmicutes/Bacteroidetes* ratio; TG, triglyceride; ALP, alkaline phosphatase; TC, total cholesterol; TBil, total bilirubin; GGT, gamma-glutamyltransferase; NAFLD, non-alcoholic fatty liver disease; FOS, fructooligosaccharides; SCFAs, short-chain fatty acids; MDA, melonaldehyde; HCC, hepatocellular carcinoma; HFrD, high-fructose diet; HSuD, high-sucrose diet; DSS, dextran sulfate sodium; CDAA, choline deficient L-amino acid defined diet; NASH, non-alcoholic steatohepatitis; HCholD, high-cholesterol diet; MSG, monosodium glutamate; s.c., subcutenoeus; LPS, lipopolysaccharides.

**Table 2 biomedicines-10-00550-t002:** Microbiota-based clinical studies in non-alcoholic fatty liver disease.

Human Study	Intervention	Outcomes	Ref.
48 patients, type-2 diabetic with NAFLD	Multi-strain probiotic mixture (*Bifidobacterium, Lactobacillus, Lactococcus, Propionibacterium*) with omega-3 fatty acids once daily for 8 weeks	Fatty liver index↓, GGT↓, TG↓, TC↓, hepatic steatosis↓, inflammatory markers↓	[177]
58 patients, type-2 diabetic with NAFLD	Multi-strain probiotic mixture (*Bifidobacterium, Lactobacillus, Lactococcus, Propionibacterium*) once daily for 8 weeks	Fatty liver index↓, GGT↓, AST↓, hepatic steatosis↓, inflammatory markers↓	[178]
64 obese children with sonographic NAFLD	Probiotic mixture (*Lactobacillus acidophilus*, *Bifidobacterium lactis*, *Bifidobacterium bifidum*, *Lactobacillus rhamnosus*)	ALT↓, AST, TG↓, TC↓, hepatic steatosis↓	[179]
39 patients with NAFLD	Multi-strain probiotic mixture (*Bifidobacterium, Lactobacillus, Streptococcus*) for 1 year	ALT↓, LPS, hepatic steatosis↓, inflammatory markers↓	[180]
102 patients with NAFLD	Synbiotic yogurt (*Bifidobacterium animalis* and Inulin) for 24 weeks	ALT↓, AST, GGT↓, ALP↓, TG↓, TC↓, fatty liver index↓, insulin resistance↓	[181]
68 obese patients with NAFLD	Probiotic mixture (*Lactobacillus*, *Pediococcus, Bifidobacterium*)	TC↓, hepatic steatosis↓, inflammatory markers↓	[182]
75 patients with NASH	Probiotic cocktail (*Lactobacillus*, *Streptococcus, Bifidobacterium*) with FOS once daily for 12 weeks and on low-fat/low-calorie diet	TC↓, ALT↓, AST, liver stiffness↓	[183]

Abbreviations: ↑ indicates an increase in the condition or level; ↓ indicates a decrease in the condition or level; ALT, alanine transaminase; AST, aspartate aminotransferase; GGT, gamma-glutamyl transferase; NAFLD, non-alcoholic liver disease; FOS, fructooligosaccharides; TG, triglyceride; ALP, alkaline phosphatase; TC, total cholesterol; LPS, lipopolysaccharides.

## 4. Limitations and Future Prospects

Due to the rapid increase in the incidence and prevalence of NAFLD and the sparse availability of effective pharmacological and therapeutic intervention, there is an urgent need to develop different approaches for the development of new drugs for treatment. Unfortunately, no targeted pharmacological approach has been approved yet. Additionally, there are no more doubts that researchers are trying to develop microbiota-based approaches for interventional therapy for NAFLD. However, considering the complexity of the gut ecosystem and the substantial microbial taxonomy variation between rodent models and human disease conditions, translating diseased rodent models of gut microbiome studies into human disease conditions are very important, as most of the bacterial species are human-specific and are yet to be determined and isolated [184]. However, multi-factor NAFLD cannot be rescinded by a generalized microbial approach, since the gut microbiome differs given their uniqueness in each host. Hence, individualized treatment approaches should be determined secondary to the common parameters that are increased in NAFLD patients. Therefore, identifying bacteria as markers based on other metabolic features and microbial communities may perhaps help in identifying patients at risk by selecting specific treatment approaches that can modulate the outcome of the treatment in NAFLD patients. Finally, the optimized manipulation of the gut microbiota could be used for microbiota-based therapeutic options for the selection of individual drug choices for better outcomes in the NAFLD progression. 

## Figures and Tables

**Figure 1 biomedicines-10-00550-f001:**
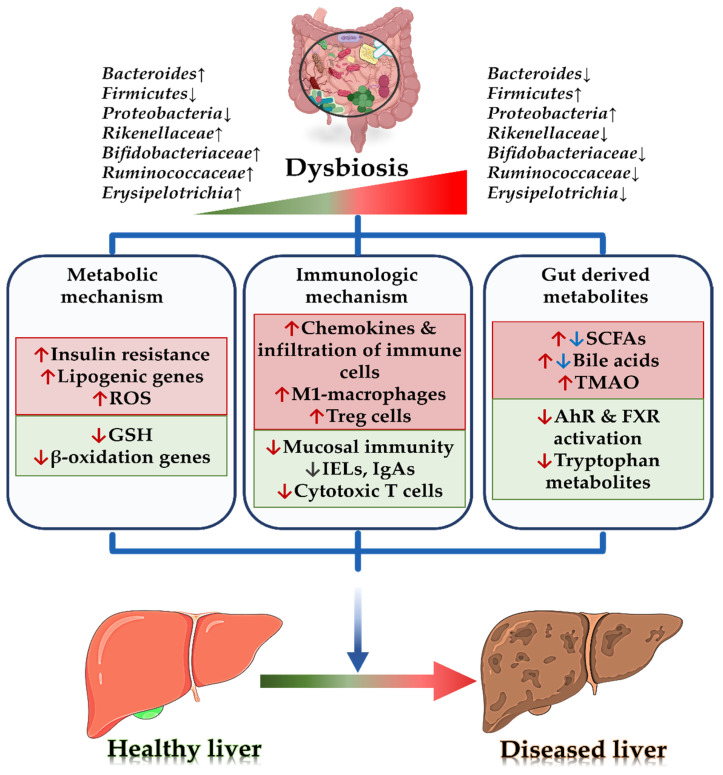
Different mechanism involving gut microbiota in pathophysiology of non-alcoholic liver disease. Abbreviations: ↑ indicates an increase in the condition or level; ROS, reactive oxygen species; GSH, glutathione; Treg, regulatory T cells; IELs, intraepithelial lymphocytes; IgAs, immunoglobulin A; SCFAs, short-chain fatty acids; TMAO, trimethylamine N-oxide; AhR, aryl hydrocarbon receptor; FXR, farnesoid X receptor.

## Data Availability

Data is contained within the article.

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
