# Peer review of "Gut Microbiome in Non-Alcoholic Fatty Liver Disease: From Mechanisms to Therapeutic Role"

_biomedicines, 2022, doi:10.3390/biomedicines10030550_

Round 1

Reviewer 1 Report

This is an interesting review analyzing, from a mechanistic point of view, the already demonstrated relationship between gut microbiome alteration and the development and progression of NAFLD.

The authors have structured the review in a comprehensible manner, and it is easy to follow their arguments. However, it is somehow disbalanced in the sense that it is too exhaustive in some respects but too short in others. For example, in point 3 of Therapeutic approaches, one would expect a little bit more on pharmacological interventions. Whether they resulted successful or not, there are many studies, trying with different drugs in NAFLD. In the same sense, the authors limit the microbiota-based interventions to pre- and pro-biotics, and one misses the references to pre-clinical experimental studies of whole microbiota transplantation or selected beneficial bacteria.

Finally, the English language needs some revision; it is very well written in most of the manuscript, but it is not in some parts of the text (as examples, the abstract, Introduction and the “De novo lipogenesis (DNL) pathway” paragraph need revision).

Author Response

biomedicines-1595423

“Gut microbiome in non-alcoholic fatty liver disease: from mechanisms to therapeutic role”

Point-to-point responses to the Reviewer’s comments

Reviewer 1:

Comment 1: This is an interesting review analyzing, from a mechanistic point of view, the already demonstrated relationship between gut microbiome alteration and the development and progression of NAFLD.

Reply: On the behalf of my team, I convey my best gratitude to the Reviewer1 for his/her comments, which helped us to express our thought in conceptual way and assisted us to improvise this manuscript relativity to the topic and readability to the general population.

Comment 2: The authors have structured the review in a comprehensible manner, and it is easy to follow their arguments. However, it is somehow disbalanced in the sense that it is too exhaustive in some respects but too short in others. For example, in point 3 of Therapeutic approaches, one would expect a little bit more on pharmacological interventions. Whether they resulted successful or not, there are many studies, trying with different drugs in NAFLD.

Reply: We are very thankful to the reviewer for this valuable and reasonable comment. Also, grateful for praising the argument flow. We are also respecting his/her view about the disbalanced in comparative arguments matter, but our primary focus of this review was only gut microbiome therapeutic advancement in NAFLD (as can be learned by the title of the articles). Moreover, our research motivation and goals are also to explore and develop microbiome-based new therapeutic avenues. Therefore, we are more concentrated and inclined towards microbiome depended on mechanism and therapies. Thus, we are requesting to the reviewer please consider our arguments flow as it is, so it can solve the main purpose and fulfill the primary goal of this review. We will really appreciate this consideration.    

Comment 3: In the same sense, the authors limit the microbiota-based interventions to pre- and pro-biotics, and one misses the references to pre-clinical experimental studies of whole microbiota transplantation or selected beneficial bacteria.

Reply: We are thankful to the reviewer for this comment and agree that we should add more microbiota-based interventions therefore we have added FMT based intervention study information starting from LINE 678-689. It also included their references. Included segment is below.

“Furthermore, modifying the microbial environment by FMT has also been attempted for NAFLD treatment. A clinical report by Vrieze et al. demonstrated that six weeks after allogenic FMT from lean donors enhanced insulin sensitivity in participants with metabolic syndrome accompanying increased levels of butyrate producing bacteria Roseburia intestinalis and Eubacterium hallii [161]. And Roseburia intestinalis was found to be negatively correlated with fibrosis and disease severity in non-obese NAFLD [84]. In addition, a recent double blinded, randomized clinical study also showed allogenic FMT from healthy donors significantly decreased the serum GGT levels and improved necrosis in liver histology in the patients with NAFLD compared to the autologous FMT. Also, significant changes were observed in the hepatic gene expression that are responsible for maintenance of endothelial integrity after allogenic FMT suggesting broadening the undiscovered horizon for the therapeutic purpose [162].”

Comment 4: Finally, the English language needs some revision; it is very well written in most of the manuscript, but it is not in some parts of the text (as examples, the abstract, Introduction and the “De novo lipogenesis (DNL) pathway” paragraph need revision).

Reply: We are very grateful to the reviewer for his/her meticulous and thorough scanning of our manuscript and very delighted to make the changes in manuscript according to his/her recommendations. We have revised and try to match as much as possible in terms of English writing flow up to our best knowledge and understandings. However, we are not the native English speakers therefore we have sent this manuscript for English editing as well. We have already submitted the English editing certificate to the editorial, the reviewer can get more information about this from editorial office. Once again thank you so much for your strong recommendation for publishing this manuscript.     

Reviewer 2 Report

In this review Authors describe microbial pathophysiology associated with non-alcoholic liver disease detailing the gut microbiota and metabolic mechanism in relation with insulin resistance, oxidative stress, and mitochondrial dysfunction. The authors also linger in describing the involvement of immune system and finally discuss the possible microbiota-based therapeutic approaches for NAFLD. I ask the authors to check the English form and the punctuation which in some places is incorrect

Author Response

Reviewer 2:

Comment 1: In this review Authors describe microbial pathophysiology associated with non-alcoholic liver disease detailing the gut microbiota and metabolic mechanism in relation with insulin resistance, oxidative stress, and mitochondrial dysfunction. The authors also linger in describing the involvement of immune system and finally discuss the possible microbiota-based therapeutic approaches for NAFLD. I ask the authors to check the English form and the punctuation which in some places is incorrect.

Reply: We convey our best gratitude to the Reviewer 2 for his/her positive, appreciative, and motivating remarks on our manuscript. We are also very delighted to make the changes in manuscript according to his/her recommendations. We have revised and try to match as much as possible in terms of English writing flow up to our best knowledge and understandings. However, we are not the native English speakers therefore we have sent this manuscript for English editing as well. We have already submitted the English editing certificate to the editorial, the reviewer can get more information about this from editorial office. Once again thank you so much for your strong recommendation for publishing this manuscript.
